# Value of endoscopic ultrasonography in the observation of the remnant pancreas after pancreatectomy

Hirotsugu Maruyama[1,2]*, Keiji Hanada[1], Akinori Shimizu[1], Tomoyuki Minami[1], Naomiti Hirano[1], Fumiaki Hino[1], Tomoyuki Abe[3], Hironobu Amano[3], Yasuhiro Fujiwara[2]

1 Department of Gastroenterology, Onomichi General Hospital, Onomichi, Japan, 2 Department of Gastroenterology, Osaka City University Graduate School of Medicine, Osaka City, Japan, 3 Department of Surgery, Onomichi General Hospital, Onomichi, Japan

* hiromaruyama99@gmail.com

**Data Availability Statement:** All relevant data are within the manuscript and its Supporting information files.

## Abstract

### Background

Endoscopic ultrasonography (EUS) is proven to be a more specific and sensitive method for detecting pancreatic lesions. However, usefulness of EUS after pancreatectomy has not been reported. This study aimed to evaluate the observational capability of EUS for the remnant pancreas (RP) after pancreatectomy.

### Patient and methods

This single-center, retrospective study enrolled 395 patients who underwent pancreatectomy at Onomichi General Hospital between December 2002 and March 2016, 45 patients who underwent EUS for RP were included for analysis. We evaluated the usefulness of EUS for RP using logistic regression analysis.

### Results

Complete observation of the RP was done in 42 patients (93%). In the initial surgical procedure, 21 patients underwent pancreaticoduodenectomy (PD), and 24 patients underwent distal pancreatectomy (DP). PD and DP were observed in 85% (18/21) and 100% (24/24) cases, respectively. A comparison of the detection capability of EUS and contrast-enhanced computed tomography (CT) or magnetic resonance imaging (MRI) showed that EUS was significantly superior to contrast-enhanced CT or MRI ($p < 0.01$). Eight of the 45 patients showed recurrence lesions in the RP. The median recurrence period was 33 months. Predictive factors for recurrence in the univariate and multivariate analyses were significantly different in space occupying lesion with EUS findings ($p < 0.01$) and elevated CA19-9 ($p < 0.01$).

### Conclusions

EUS was able to observe the RP in almost all cases. In addition, the detection capability of EUS was significantly superior to those of CT or MRI. We recommend that all patients with RP should undergo EUS, and a longer follow-up must be performed.

**Funding:** The author(s) received no specific funding for this work.

**Competing interests:** The authors have declared that no competing interests exist.

## Introduction

Pancreatic ductal adenocarcinoma, simply termed pancreatic cancer (PC) in the review, is one of the most lethal malignancies and has a poor prognosis with an overall 5-year survival rate of approximately 5% [1,2]. Most patients initially presented with clinically advanced disease, and only 10–15% were candidates for surgical resection. Even among patients who underwent surgery with curative-intent, more than 90% developed disease progression within 12–18 months [3]. The major sites of recurrence are the local pancreatic bed, liver and the peritoneal surface. However, Miyazaki et al. reported that 11 (3.9%) of 284 cases with pancreatectomy had recurrence in the remnant pancreas (RP) [4]. Because metachronous PC can be treated with surgery [5], observation of the RP and regular follow-up are important.

National Comprehensive Cancer Network (NCCN) guidelines recommend the surveillance of serum carbohydrate antigen (CA) 19–9 levels and computed tomography (CT) examination every 3 to 6 months for 2 years after pancreatectomy [6]. The postoperative CT findings of patients with PC are generally liver metastasis, lymph node metastasis and local recurrence (nerve and vascular invasion) [7]. This result suggests that CT alone may not detect small lesions in the RP. Ikemoto et al. reported that RP could be observed using endoscopic ultrasonography (EUS). Thus, regular follow-up was recommended as well for pancreatic disease [8]. However, to our knowledge, no study has reported the usefulness of EUS for RP.

One of the most promising techniques for early detection of pancreatic lesions is EUS. EUS has been considered an essential tool for diagnosing PC and assessing the extent and resectability of pancreatic tumors [9]. In addition, follow-up EUS reportedly improves lesion detection compared with multidetector CT alone [10]. Therefore, we believe that intervention with EUS as well as CT examination is necessary for postoperative follow-up.

This study aimed to retrospectively investigate the evaluation of the observational capability of EUS for RP after pancreatectomy.

## Materials and methods

### Patients/Material

We retrospectively analyzed consecutive patients who underwent pancreatectomy for PC and intraductal papillary mucinous neoplasm (IPMN) at the Onomichi General Hospital. We enrolled 395 patients between December 2002 and March 2016. Among 395 patients, 45 who underwent EUS for RP were included. Patients aged >20 years and those who underwent EUS after pancreatectomy were also included. In contrast, those who did not have a detailed record were excluded.

### Ethical consideration

The ethics committee of the Onomichi General Hospital approved the study protocol; (number 2019–13), which waived the requirement for written informed consent because the analysis used anonymized clinical data that were retrospectively obtained after each patient agreed to receive the treatment. Nevertheless, all patients were notified of the content and information of this study and given the opportunity to refuse participation. None of the patients refused participation. This study followed the Ethical Guidelines for Medical and Health Research Involving Human Subjects established by the Ministry of Education, Culture, Sports, Science and Technology and the Ministry of Health, Labor and Welfare in Japan.

### Main outcome measurements

Evaluation of the observational capability of EUS for RP after pancreatectomy.

## Initial surgical procedure

In the initial surgical procedure, pancreaticoduodenectomy (PD) including pylorus-preserving PD (PPPD) and subtotal stomach-preserving PD (SSPPD) were constructed using PD-IIA (Child method) in all cases. In distal pancreatectomy (DP), the pancreatic parenchyma was divided by a linear stapler.

## Follow—Up strategy after pancreatectomy

We performed a blood test, along with the determination of CA19-9 and carcinoembryonic antigen (CEA) levels every 3 to 6 months and contrast-enhanced CT or magnetic resonance imaging (MRI) every 6 to 12 months. EUS was performed if contrast-enhanced CT or MRI revealed abnormal findings or elevated levels of some tumor markers.

## EUS procedure

EUS was performed using a radial array echoendoscope (GF-UM2000 and GF-UE260, Olympus Medical Systems, Tokyo, Japan) equipped with a processor (UE ME-1 and UE ME-2, Olympus Medical Systems, Tokyo, Japan). EUS-guided fine-needle aspiration (EUS-FNA) was performed using a linear array echoendoscope (GF-UCT260, Olympus Medical Systems, Tokyo, Japan). EUS and EUS-FNA were performed by an endoscopist with more than 5 years of experience. Linear array EUS was used selectively only in the case of EUS-FNA.

**PD case.** PD cases were performed using the transgastric approach. The body and tail of the pancreas were continuously observed through confirmation of neighboring organs and the splenic vein (SPV) and superior mesenteric artery. Advancement of the echoendoscope along the SPV demonstrated the anastomotic part (liner high echo) and body of the pancreas by the confluence (Fig 1a). The "linear high echo" represents the digestive tract wall.

**DP case.** We confirmed the liner high echo in the transgastric approach. Observation from the bulb of the duodenum provided images of the bile duct, confluence, head of the pancreas and part of the pancreas body. Observation from the descending part of the duodenum provided images of the head of the pancreas and the ampulla (Fig 1b). The "linear high echo" represents the stapler.

**Evaluation of complete observation by EUS for the RP.** The evaluation of complete observation of the pancreas by EUS was performed by a board-certified fellow of the Japan Gastroenterological Endoscopy Society, with > 8 years of experience. All patients were complete with; endoscopic images, reports, and videos. The location (head, body and tail of the pancreas), background of the pancreas (chronic pancreatitis, atrophy, etc.), lesion size, number, echotexture (homogeneous, heterogeneous), echogenicity (hyperechoic, hypoechoic, anechoic), operation methods, anastomotic part, site of the difficult observation and diagnosis were described in the endoscopic reports. Along with checking the endoscopic report findings, all endoscopic images were checked. The videos were referred to when the endoscopic images could not be used for analysis. The complete observation of the pancreas was evaluated based on its similarity with that of a normal organ. There was no difficult observation.

## Imaging acquisition

Contrast-enhanced CT examination was performed using multi-detector CT machines. Arterial phase scanning began 35–40 seconds after injection of 2ml/kg of body weight of a nonionic iodinated contrast agent at a rate of 4ml/s with a bolus-triggered technique using an automatic power injector. Portal and delayed phase scanning began 70 and 180 seconds after the start of the contrast medium injection, respectively. The slice thickness was 2 or 5-mm.

## a) Pancreaticoduodenectomy case

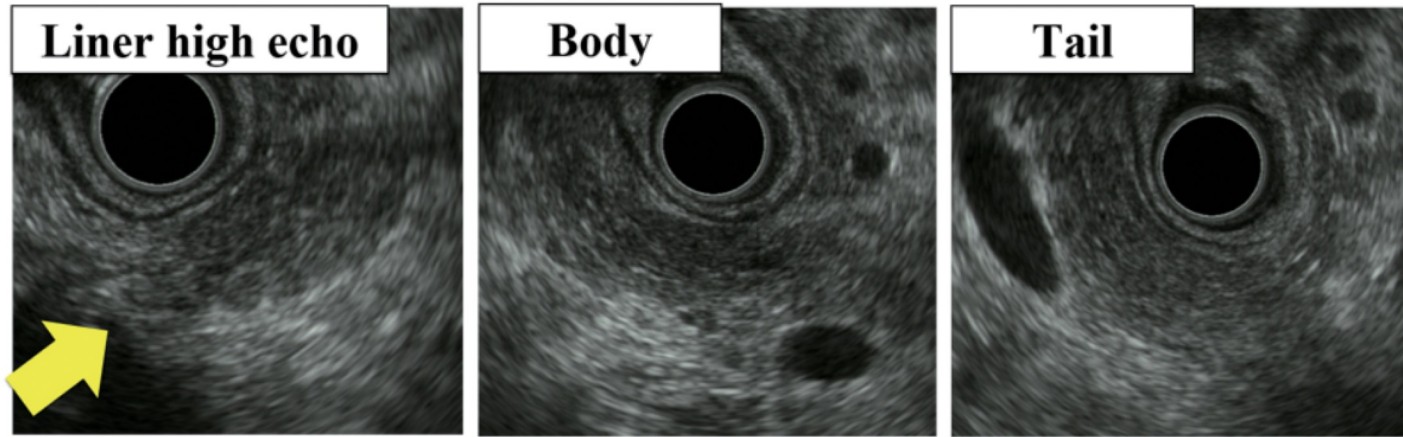

## b) Distal pancreatectomy cases

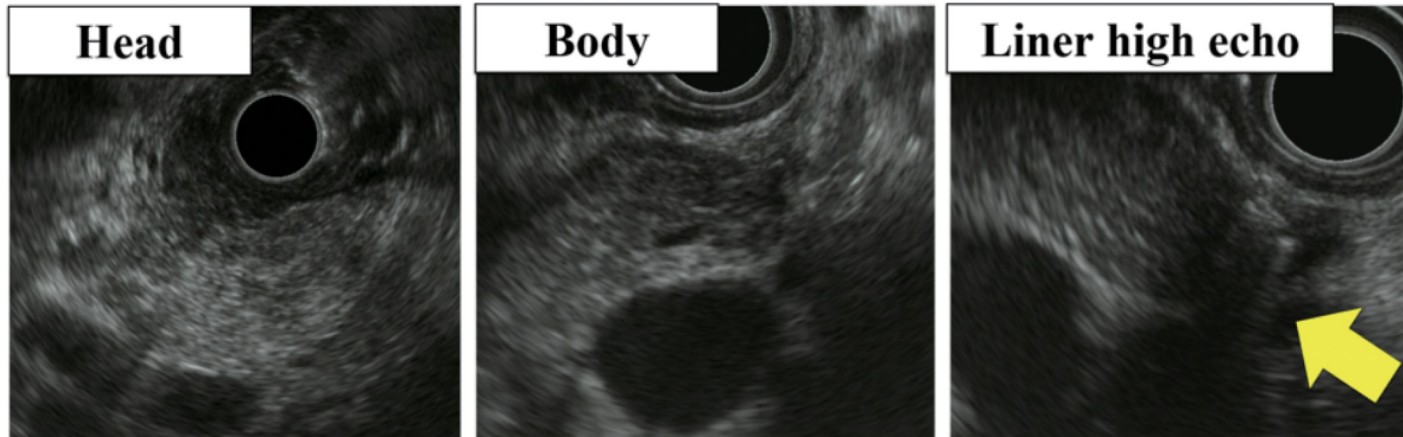

**Fig 1. Observation of endoscopic ultrasonography (EUS) for the remnant pancreas.** a) We showed the EUS images of each parts in pancreaticoduodenectomy case. b) We showed the EUS images of each parts in distal pancreatectomy case.

MRI examination was performed using a 1.5 and 3.0 Tesla system. We used a 3.0 Tesla system from May 2011. MRI images were acquired using the following sequences: a T1 weighted sequence (in phase and opposed phase), T2 weighted sequence, FatSat sequence, diffusion weighted sequence, and magnetic resonance cholangiopancreatography.

**Evaluation of the contrast-enhanced CT and MRI findings.** Two radiologists reviewed the contrast-enhanced CT and MRI images. In addition, it was checked contrast-enhanced CT and MRI imaging by board certified fellow of the Japan Gastroenterological Endoscopy Society who has more than 8 years of experience in EUS.

## Data collection

All data were extracted from paper based and electronic medical records. We entered the results of all consecutive EUS attempts during the study period and retrospectively recorded the observation of the RP, detection of the lesion in the RP, and patient- and procedure-related data on a detailed data collection sheet.

## Statistical analysis

To summarize the patients' baseline clinical and demographical characteristics, medians and interquartile ranges were used for continuous variables and percentages and counts were used for categorical variables. The statistical analysis was performed using either two-sided $\chi^2$ test, depending on the characteristics of the data. Binary logistic regression analysis was used to evaluate the predictive factors for recurrence, estimated by calculating the odds ratios (ORs) and the 95% confidence intervals (CIs). A $P$ value of <0.05 was considered significant. All statistical analysis was conducted using the R software (version 3.3.2, F Foundation for Statistical Computing, Vienna, Austria) using the "rms" package.

# Results

## Baseline characteristics of patients

The median age was 65 years (interquartile range; 61–72 years). With respect to the initial surgical procedure, 21 patients underwent PD, while 24 underwent DP. Thirty-three patients had PC, two of them had PC concomitant with IPMN of the pancreas. The other patients were 10 cases of IPMN and 2 cases of Intraductal Papillary Mucinous Carcinoma (IPMC). The pathological stages of 33 cases of PC were as follows; stage 0, 11; stage Ia, 8; stage Ib, 2; and stage IIb, 12. Fourteen of the 45 cases showed recurrence detected in the RP (n = 8), liver metastasis (n = 4), lymph node metastasis (n = 2), local nerve invasion (n = 1), and peritoneal dissemination (n = 2); there were overlapping sites of recurrence. The patients' characteristics are shown in Table 1.

## Evaluation of the observational capability of EUS for the RP

The complete observation rate of the RP was approximately 93% (42/45). Complete observation of the patients who underwent DP was 100% (24/24), however, it of the cases who

**Table 1. Patients characteristics.**

| | |
|---|---|
| No. of patients | 45 |
| Age, median (IQR), years | 65 (61–72) |
| Male / female, (%) | 25/20 (56/44) |
| Initial surgical procedure | |
| Pancreaticoduodenectomy (PD) cases | 21 |
| Pylorus-preserving PD cases | 2 |
| Subtotal stomach-preserving PD cases | 18 |
| Pancreaticoduodenectomy cases | 1 |
| Distal pancreatectomy cases | 24 |
| IPMN / IPMC / Pancreatic cancer (PC) cases | 10 / 2 / 33 |
| PC fstage 0 / Ia / Ib / IIb cases | 12 / 8 / 2 / 11 |
| Preoperative CA19-9, median (IQR), ng/ml | 13 (5.3–53) |
| Recurrence cases, n (%) | 14 (31) |
| Remnant pancreas | 8 (17.7) |
| Liver metastasis | 4 (8.8) |
| Lymph node metastasis | 2 (4.5) |
| Local (nerve invasion) | 1 (2.2) |
| Peritoneal dissemination | 2 (4.5) |

Table 1 shows the clinical characteristics of the patients.

IQR: Interquartile range. IPMN: Intraductal papillary mucinous neoplasm. IPMC: Intraductal papillary mucinous carcinoma. PC: Pancreatic cancer, CA19-9: Carbohydrate antigen 19–9.

**Table 2. The results of pancreaticoduodenectomy and distal pancreatectomy cases.**

| | Pancreaticoduodenectomy cases n = 21 | Distal pancreatectomy cases n = 24 |
|---|---|---|
| Age, median (IQR), years | 62 (57–70) | 66.5 (63–74) |
| Male / female | 16/ 5 | 9/ 15 |
| IPMN/IPMC/PC cases (n) | 5/ 2/ 14 | 5/ 0/ 19 |
| PC pstage 0/ Ia/ Ib/ IIb cases (n) | 3/ 5/ 1/ 5 | 9/ 3/ 1/ 6 |
| CA19-9, median (IQR), ng/ml | 15 (2–77.2) | 10.4 (6.4–53.4) |
| Recurrence cases, n (%) | | |
| Remnant pancreas | 5 (23.8) | 3 (12.5) |
| Complete observation of RP by EUS, n (%) | 18 (85) | 24 (100) |
| Detection of abnormal findings in RP, n (%) | | |
| EUS examination | 14 (66.7) | 13 (54.2) |
| Contrast-enhanced CT or MRI examination | 10 (47.6) | 6 (25) |

Table 2 shows the results of pancreaticoduodenectomy and distal pancreatectomy case.

IQR: Interquartile range. IPMN: Intraductal papillary mucinous neoplasm. IPMC: Intraductal papillary mucinous carcinoma, PC: Pancreatic cancer, CA19-9: Carbohydrate antigen 19–9, RP: Remnant pancreas, EUS: Endoscopic ultrasonography, CT: Computed tomography, MRI: Magnetic resonance images.

underwent DP was 85% (18/21). The anastomotic part could not be observed in three cases of PD. The results of the PD and DP cases are shown in Table 2.

## Evaluation of the capability of EUS, contrast-enhanced CT or MRI to detect abnormal findings in the RP

The findings of patients who underwent EUS were as follows: 18 cases had no findings, 11 had cysts, 12 had a space occupying lesion (SOL), 4 had main pancreatic duct (MPD) dilatation, and 3 had chronic pancreatitis (CP). The findings of patients who underwent contrast-enhanced CT or MRI were as follows: 29 cases had no findings, 11 had cysts, 3 had SOL, 4 had MPD dilatation, and none of them had CP (both EUS and contrast-enhanced CT or MRI findings overlapped). Among the 45 patients, 17 had no remarkable findings on both EUS and contrast-enhanced CT or MRI. Twelve cases were only EUS findings, and 1 case was only contrast-enhanced CT or MRI findings (Table 3). A comparison of the detection capability of EUS and contrast-enhanced CT or MRI showed that EUS was significantly superior to contrast-enhanced CT or MRI ($p < 0.01$).

**Table 3. Abnormal findings in the remnant pancreas.**

| Findings | EUS (n) | CT or MRI (n) |
|---|---|---|
| Total detection cases | 27 | 16 |
| Content | | |
| No findings | 18 | 29 |
| Cyst | 11 | 11 |
| SOL | 12 | 4 |
| MPD dilatation | 4 | 5 |
| CP | 3 | 0 |

Table 3 shows the abnormal findings in the remnant pancreas.

EUS: Endoscopic ultrasonography. CT: Computed tomography. MRI: Magnetic resonance images. SOL: Space occupying lesion. MPD dilatation: Main pancreatic duct dilatation. CP: Chronic pancreatitis.

**Table 4. Univariate and multivariate binary logistic regression analysis of PEP.**

| Variables | | No recurrence (n = 31) | Recurrence (n = 14) | Crude OR (95% CI) | P value | Multivariate OR (95% CI) | P value |
|---|---|---|---|---|---|---|---|
| Age, median (IQR) | | 65 (61.5–71) | 66 (59.5–72) | 0.98 (0.90–1.05) | 0.51 | | |
| Sex | | | | 8.3 (1.58–43.6) | 0.01 | | |
| Male | | 13 | 12 | | | | |
| Female | | 18 | 2 | | | | |
| CA19-9, IU/mL | | | | 12.6 (2.72–58.0) | < 0.01 | 24.7 (2.36–259) | < 0.01 |
| ≦37 | | 25 | 3 | | | | |
| > 37 | | 6 | 11 | | | | |
| **Contrast-enhanced CT or MRI examination findings (pancreas)** | | | | | | | |
| Cyst | yes | 10 | 1 | 0.16 (0.019–1.41) | 0.1 | | |
| main pancreatic duct dilatation | yes | 3 | 2 | 1.56 (0.23–10.5) | 0.65 | | |
| low density area | yes | 1 | 3 | 8.18 (0.77–87.2) | 0.082 | 0.34 (0.01–12.4) | 0.56 |
| **Endoscopic ultrasonography examination findings (pancreas)** | | | | | | | |
| Cyst | yes | 10 | 1 | 0.16 (0.019–1.41) | 0.1 | | |
| main pancreatic duct dilatation | yes | 3 | 1 | 0.72 (0.068–7.58) | 0.78 | | |
| space occupying lesion | yes | 3 | 9 | 16.8 (3.34–84.6) | < 0.01 | 42.2 (2.8–636) | < 0.01 |

Table 4 shows the univariate and multivariate analysis of PEP.

IQR: Interquartile range. CT: computed tomography. MRI: Magnetic resonance images.

Twenty-one patients underwent EUS solely due to CA19-9 elevation. Among 21 cases, 8 had no findings on both EUS and contrast-enhanced CT or MRI. Thirteen patients showed some abnormalities by EUS, and 7 showed contrast-enhanced CT or MRI findings. There were no abnormal findings on contrast-enhanced CT or MRI alone. A comparison of the detection capability of EUS and contrast-enhanced CT or MRI showed that EUS was significantly superior to contrast-enhanced CT or MRI ($p = 0.02$). Among the 21 cases, 8 cases underwent EUS-FNA. In 5 of these cases, EUS changed the diagnostic algorithm.

Furthermore, univariate and multivariate analyses for recurrence are shown in Table 4. Predictive factors for recurrence in the univariate and multivariate analyses were significantly different in SOL on EUS findings (OR; 42.2, 95%CI; 2.8–636, $p < 0.01$) and elevated CA19-9 (OR; 24.7, 95%CI; 2.36–259, $p < 0.01$).

## Cases with PC recurrence in the RP

Eight of the 45 cases showed recurrence lesions in the RP (Table 5). Seven patients had PC, and 1 had IPMN. The preoperative stages of patients who required PC were as follows: 1 case had stage 0, 3 had stage Ia, and 3 had stage IIb. The median recurrence period was 33 months, and the longest recurrence period was 84 months. Although contrast-enhanced CT or MRI was able to detect it in only 3 cases, EUS was able to detect it in the RP for all cases. Therefore, EUS-FNA was performed in all cases to detect the presence of lesions in the RP. Six cases showed positive pathological results. One of the other two cases showed atypical lesions; however, it was diagnosed with recurrence after surgery. The other case was strongly suspected to have a recurrence based on the positron emission tomography findings. A second pancreatectomy was performed in 5 of 8 cases. EUS-FNA had a sensitivity of 75% (6/8) and an accuracy of 75% (6/8).

**Table 5. Cases with pancreatic cancer recurrence in the remnant pancreas.**

| Case | Preoperative stage (UICC 7th) | Surgical procedure | Histological findings | Residual tumor | Recurrence period (month) | EUS findings | CT/MRI findings | FNA (G) | Other modality | Second surgery |
|------|------|------|------|------|------|------|------|------|------|------|
| 1. 74 / F | Ia | DP | Well differentiated | R0 | 44 | + | + | Positive (25) | - | + |
| 2. 57 / M | - | SSPPD-IIA-2 | IPMN | - | 19 | + | - | Positive (25) | - | - |
| 3. 72 / M | IIb | SSPPD-IIA-2 | Poorly differentiated | R0 | 12 | + | + | Positive (22) | - | + |
| 4. 62 / M | IIb | SSPPD-IIA-2 | Moderately differentiated | R0 | 22 | + | - | Positive (25) | - | - |
| 5. 61 / M | IIb | DP | Papillary | R0 | 16 | + | - | Negative (25) | PET | - |
| 6. 70 / M | 0 | DP | Well differentiated | R0 | 74 | + | - | Atypical (25) | - | + |
| 7. 75 / F | Ia | PD-IIA-2 | Poorly differentiated | R0 | 84 | + | - | Positive (25) | - | + |
| 8. 39 / M | Ia | PD-IIA-2 | Papillary | R0 | 62 | + | + | Positive (25) | - | + |

Table 5 shows cases with pancreatic cancer recurrence in the remnant pancreas.

M: Male. F: Female. DP: Distal pancreatectomy. SSPPD: Subtotal stomach-preserving pancreaticoduodenectomy. PD: Pancreaticoduodenectomy. R: Residual tumor. EUS: Endoscopic ultrasonography. CT: Computed tomography. MRI: Magnetic resonance images. FNA: Fine-needle aspiration. PET: Positron emission tomography.

## Discussion

Our findings found that EUS had a greater capability to observe RP in almost all cases due to several reasons. First, radial array echoendoscope was used which produce US images perpendicular to the axis of the endoscope tip with a 360-degree scanning range, thus providing circumferential images and hence a lot of information to the endoscopist. Therefore, these were easy to interpret even for the pancreatectomy cases. Second, all PD at our hospital were performed using the Child reconstruction (PD-IIA) method, in which the RP is most likely located on the dorsal side of the stomach. Therefore, it was possible to observe the RP using a radial array echoendoscope.

Our results showed that the anastomotic part could not be observed in 3 PD cases. For these reasons, the anastomotic part was located far from the stomach in the PD-IIA method. EUS was used in the observation of RP [11,12]. However, these studies did not include the evaluation of its observation capability of EUS.

In our study, the detection capability of EUS and contrast-enhanced CT or MRI was compared. Abnormal findings were more frequently detected on EUS. In particular, EUS had a greater capability to detect any SOL in the RP. Second, we investigated CA19-9 elevation because some patients were enrolled due to positive findings on contrast-enhanced CT or MRI. A comparison of the detection capability of EUS and contrast-enhanced CT or MRI showed that EUS was significantly superior to contrast-enhanced CT or MRI ($p = 0.02$). In addition, predictive factors for recurrence in the univariate and multivariate analyses indicated a significant difference with SOL in EUS findings and elevated CA19-9. In the clinical course, EUS changed the diagnostic algorithm in 5 cases with CA19-9 elevation. Therefore, we consider CA19-9 elevation and the SOL finding of EUS to be useful parameters in recurrence of the remnant pancreas. As reported in previously studies, EUS remained the best imaging modality for detecting pancreatic lesions compared to CT and MRI [13]. The original study published by the Cancer of the Pancreas Screening-3 comparing the findings of CT, MRI, and EUS showed that EUS had the highest detection rate for pancreatic lesions with 36% diagnostic yield, while MRI and CT only showed a diagnostic yield of 33% and 11%, respectively [14]. These results support the findings of our study. We believe that EUS is the best imaging modality even for the observation of an RP.

In our study, 8 of the 45 patients showed recurrent lesions in the RP. None of the patients had residual tumors after initial surgery. Recurrence in the RP was observed in three patients

who underwent contrast-enhanced CT or MRI examination, whereas EUS enabled observation in the RP of all patients. EUS-FNA was performed in all patients to detect lesions in the RP, and a second pancreatectomy was performed in five out of eight patients. Therefore, EUS-FNA may be useful in the pathological diagnosis of PC in the RP [12]. With respect to the recurrence period, four patients had recurrence within 2 years after the initial surgery. However, the other patients had a recurrence period of more than 2 years. Three patients had recurrence over 5 years, and the longest recurrence period was 7 years. Therefore, we believe that longer follow-up is necessary rather than two years, as recommended by the NCCN guidelines.

The major strength of our study is that it is the first, to our knowledge, to investigate the capability of EUS to observe RP after pancreatectomy. Second, this study included a long-term follow-up of patients with PC or IPMN postoperatively. Third, we found that EUS is superior to contrast-enhanced CT and MRI in terms of detecting SOLs in the RP.

Our study has several limitations. First, the analyses were based on retrospectively collected data. Second, the variability in echoendoscopes and processors may change the capability to detect pancreatic lesions. Third, we did not compare pre-operative and post-operative EUS because, 13 patients did not receive pre-operative EUS and all patients had passed several years since pre-operative EUS. Therefore, the condition of the background pancreas had changed and accurate comparisons were difficult.

Even though patients may benefit from curative resection, high rates of recurrence are still present, including liver metastasis recurrence, local recurrence, lymph node recurrence, and peritoneal dissemination within 2 years after surgery. However, the intervention of diagnostic imaging including EUS, if contrast-enhanced CT or MRI revealed abnormal findings or some tumor marker levels were elevated may improve the treatment outcome. We believe that EUS intervention with recurrence of the remnant pancreas in mind will improve long-term prognosis, with regard to post-operative cases of PC.

In conclusion, EUS enables observation of the RP in almost all cases. In addition, the detection capability of EUS was significantly superior to that of contrast-enhanced CT or MRI.

## Supporting information

**S1 File.**
(CSV)

## Acknowledgments

Dr. Maruyama would like to express his deepest gratitude to Professor Kazuki Chayama of the Department of Gastroenterology and Metabolism of Hiroshima University for the cooperation in training at Onomichi General Hospital. A part of this report was presented at the 25th United European Gastroenterology Week (Barcelona, Spain, October 2017). Authors declare no conflict of interest for this article.

## Author Contributions

**Conceptualization:** Hirotsugu Maruyama, Keiji Hanada.

**Data curation:** Hirotsugu Maruyama.

**Formal analysis:** Hirotsugu Maruyama, Keiji Hanada.

**Methodology:** Hirotsugu Maruyama, Keiji Hanada.

**Resources:** Yasuhiro Fujiwara.

**Supervision:** Keiji Hanada.

**Writing – original draft:** Hirotsugu Maruyama.

**Writing – review & editing:** Akinori Shimizu, Tomoyuki Minami, Naomiti Hirano, Fumiaki Hino, Tomoyuki Abe, Hironobu Amano, Yasuhiro Fujiwara.

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
