## [Decision Letter · Decision Letter 0]

13 Oct 2020

PONE-D-20-25624

Value of endoscopic ultrasonography for the remnant pancreas

PLOS ONE

Dear Dr. Maruyama,

Thank you for submitting your manuscript to PLOS ONE. After careful consideration, we feel that it has merit but does not fully meet PLOS ONE’s publication criteria as it currently stands. Therefore, we invite you to submit a revised version of the manuscript that addresses the points raised during the review process.

This study did not receive the approval from the Reviewers. See the comments below

We look forward to receiving your revised manuscript.

Kind regards,

Roberto Coppola, MD, FACS

Academic Editor

PLOS ONE

Journal Requirements:

2. In ethics statement in the manuscript and in the online submission form, please provide additional information about the patient records/samples used in your retrospective study. Specifically, please ensure that you have discussed whether all data/samples were fully anonymized before you accessed them and/or whether the IRB or ethics committee waived the requirement for informed consent. If patients provided informed written consent to have data/samples from their medical records used in research, please include this information."

3. To comply with PLOS ONE submission guidelines, in your Methods section, please provide additional information regarding your statistical analyses. For more information on PLOS ONE's expectations for statistical reporting, please see https://journals.plos.org/plosone/s/submission-guidelines.#loc-statistical-reporting.”

Additional Editor Comments (if provided):

I appreciate the work done by the Authors. The topic is of interest, however I would add some comments and questions:

- the Authors should state if PC patients received EUS before Surgery and if data were compared with those achieved during the EUS performed on RP

- did the AUthors performe pancreatic margin frozen section examination? If they did, they shoud state at least how many IPMN have been found in the 45 cases object of the study

- text and English Language should be revised

Reviewers' comments:

Reviewer's Responses to Questions

**Comments to the Author**

1. Is the manuscript technically sound, and do the data support the conclusions?

Reviewer #1: Partly

2. Has the statistical analysis been performed appropriately and rigorously? 

Reviewer #1: I Don't Know

3. Have the authors made all data underlying the findings in their manuscript fully available?

Reviewer #1: Yes

4. Is the manuscript presented in an intelligible fashion and written in standard English?

Reviewer #1: No

5. Review Comments to the Author

Reviewer #1: I appreciate the work done by the Authors. The topic is of interest, however I would add some comments and questions:

- the Authors should state if PC patients received EUS before Surgery and if data were compared with those achieved during the EUS performed on RP

- did the AUthors performe pancreatic margin frozen section examination? If they did, they shoud state at least how many IPMN have been found in the 45 cases object of the study

- text and English Language should be revised

6. PLOS authors have the option to publish the peer review history of their article (what does this mean?). If published, this will include your full peer review and any attached files.

Reviewer #1: No

---

## [Author Response · Author response to Decision Letter 0]

10 Dec 2020

Joerg Heber

Editor-in-Chief

Associate Editors

PLOS ONE

Dear Editor

We appreciate the opportunity to submit a revised version of our manuscript entitled, “Value of endoscopic ultrasonography in the observation of the remnant pancreas after pancreatectomy” (manuscript ID: PONE-D-20-25624) for consideration for publication in PLOS ONE. 

We believe that we have addressed all concerns raised by the reviewers as detailed in the accompanying point-by-point responses. 

All authors concur with the submission of this manuscript. We ascertain that none of the data in this manuscript have been previously reported, nor is the manuscript under consideration for publication elsewhere.

We hope that PLOS ONE now finds our manuscript suitable for publication. We appreciate your consideration of our work.

Sincerely Yours,

Hirotsugu Maruyama

Hirotsugu Maruyama, M.D., PhD.

Department of Gastroenterology

Osaka City University Graduate School of Medicine 

1-4-3, Asahimachi, Abeno-ku, Osaka-City, Osaka, 545-8585, Japan

e-mail to; hiromaruyama99@gmail.com

Phone: +81-6-6645-3811

FAX: +81-6-6645-3813

 

Point-by-Point Responses (PONE-D-20-25624)

Responses to the Editorial and Reviewer comments:

We appreciate the editor’s positive and helpful comments about our paper. The reviewer raised some important points for improvement that we have now addressed, as summarized below. Please note that all changes are yellow highlights in the revised manuscript.

Comment 1:

- the Authors should state if PC patients received EUS before Surgery and if data were compared with those achieved during the EUS performed on RP. 

Response 1:

Thank you for your constructive comment. We could not compare preoperative and postoperative EUS, because of several limitations. First, 32 patients received pre- and post-operative EUS; however, 13 patients did not receive preoperative EUS. Second, it is difficult to compare pre- and post-operative EUS because all patients have been several years since preoperative EUS and the condition of the background pancreas has changed. 

We described it in the DISCUSSION (limitation): " Our study has several limitations. First, the analyses were based on retrospectively collected data. Second, the variability in echoendoscopes and processors may change the capability to detect pancreatic lesions. Third, we did not compare pre-operative and post-operative EUS because, 13 patients did not receive pre-operative EUS and all patients had passed several years since pre-operative EUS. Therefore, the condition of the background pancreas had changed and accurate comparisons were difficult." (from page 22, line 4 to 7)

Comment 2:

- did the AUthors performe pancreatic margin frozen section examination? If they did, they shoud state at least how many IPMN have been found in the 45 cases object of the study.

Response 2:

Thank you for your constructive comment. All cases underwent pancreatic margin frozen section examination. Among the 45 cases, 14 were IPMN cases. In the text, we described "Thirty-third patients had PC. The other patients were 10 cases of IPMN and 2 cases of Intraductal Papillary Mucinous Carcinoma (IPMC)". Thus, 12 cases were described in the text. However, among the 33 cases of pancreatic cancer, 2 cases were pancreatic ductal adenocarcinoma concomitant with IPMN of the pancreas.

We described in the Results (Baseline characteristics of patients): " Thirty-three patients had PC, two of them had PC concomitant with IPMN of the pancreas. The other patients were 10 cases of IPMN and 2 cases of Intraductal Papillary Mucinous Carcinoma (IPMC)." (from page 12, line 8 to 11)

Comment 3:

- text and English Language should be revised.

Response 3:

We appreciate your comment.

Our manuscript had undergone English proofreading by a native speaker, and has been revised once again by a professional English language editing service; the calibration certificate has been attached. Please see the revised manuscript.

---

## [Editor Report · Decision Letter 1]

2 Jan 2021

Value of endoscopic ultrasonography in the observation of the remnant pancreas after pancreatectomy

PONE-D-20-25624R1

Dear Dr. Maruyama,

We’re pleased to inform you that your manuscript has been judged scientifically suitable for publication and will be formally accepted for publication once it meets all outstanding technical requirements.

Kind regards,

Roberto Coppola, MD, FACS

Academic Editor

PLOS ONE

Additional Editor Comments (optional):

The changes made by the Authors in the revised manuscript are complete. The document can be accepted for publication.
---

## [Editor Report · Acceptance letter]

8 Jan 2021

PONE-D-20-25624R1 

Value of endoscopic ultrasonography in the observation of the remnant pancreas after pancreatectomy 

Dear Dr. Maruyama:

I'm pleased to inform you that your manuscript has been deemed suitable for publication in PLOS ONE. Congratulations! Your manuscript is now with our production department. 

Kind regards, 

on behalf of

Professor Roberto Coppola 

Academic Editor

PLOS ONE